# Association of Genomic Alterations with the Presence of Serum Monoclonal Proteins in Chronic Lymphocytic Leukemia

**DOI:** 10.3390/cells13221839

**Published:** 2024-11-07

**Authors:** Juan A. Piñeyroa, Irene López-Oreja, Ferran Nadeu, Ares Martínez-Farran, Juan Ignacio Aróstegui, Mónica López-Guerra, Juan Gonzalo Correa, Aleix Fabregat, Neus Villamor, Ines Monge-Escatín, Nil Albiol, Dolors Costa, Marta Aymerich, Sílvia Beà, Elías Campo, Julio Delgado, Dolors Colomer, Pablo Mozas

**Affiliations:** 1Department of Hematology, Hospital Clínic, 08036 Barcelona, Spain; japineyroa@gmail.com (J.A.P.); jgonzalo@clinic.cat (J.G.C.); nalbiol@clinic.cat (N.A.); jdelgado@clinic.cat (J.D.); 2Institut d’Investigacions Biomèdiques August Pi i Sunyer (IDIBAPS), 08036 Barcelona, Spain; ilopezor@clinic.cat (I.L.-O.); nadeu@recerca.clinic.cat (F.N.); armartinez@recerca.clinic.cat (A.M.-F.); jiaroste@clinic.cat (J.I.A.); lopez5@clinic.cat (M.L.-G.); afabregat@clinic.cat (A.F.); villamor@clinic.cat (N.V.); dcosta@clinic.cat (D.C.); aymerich@clinic.cat (M.A.); sbea@clinic.cat (S.B.); ecampo@clinic.cat (E.C.); 3Facultat de Medicina i Ciènces de la Salut, Universitat de Barcelona, 08036 Barcelona, Spain; 4Centro de Investigación Biomédica en Red de Cáncer (CIBERONC), 28029 Madrid, Spain; 5Hematopathology Unit, Department of Pathology, Hospital Clínic, 08036 Barcelona, Spain; 6Department of Immunology, Hospital Clínic, 08036 Barcelona, Spain; 7Department of Biochemistry and Molecular Genetics, Hospital Clínic, 08036 Barcelona, Spain; 8Pharmacy Unit, Hospital Clínic de Barcelona, 08036 Barcelona, Spain; monge@clinic.cat

**Keywords:** chronic lymphocytic leukemia, serum monoclonal protein, next-generation sequencing

## Abstract

The presence of a monoclonal protein detected by serum immunofixation electrophoresis (sIFE) has been reported as an adverse prognostic factor in chronic lymphocytic leukemia (CLL). However, the genetic underpinning of this finding has not been studied. We retrospectively studied 97 CLL patients with simultaneous information on sIFE and genetic alterations detected by next-generation sequencing. sIFE was positive in 49 patients. The most common isotypes were IgG κ (27%) and bi/triclonal (25%). A +sIFE was associated with a higher number of mutated genes [median 2 (range 0–3) vs. 0 (range 0–2), *p* = 0.006], and a higher frequency of unmutated IGHV status (60 vs. 29%, *p* = 0.004). An IgM monoclonal protein was associated with *TP53* mutations (36% in IgM +sIFE vs. 12% in non-IgM +sIFE or –sIFE, *p* = 0.04), and bi/triclonal proteins with *NOTCH1* mutations (33% in bi/triclonal vs. 9% in monoclonal +sIFE or –sIFE, *p* = 0.04). These data suggest an association between a +sIFE and a higher mutational burden, and some monoclonal isotypes with specific mutations.

## 1. Introduction

Immunoglobulin secretion is a function of terminally differentiated B-cells. These B-cells are released into the periphery as plasma cells or long-lived memory B-cells and can secrete a monoclonal immunoglobulin composed of a heavy and a light chain [1]. The presence of serum monoclonal proteins is usually associated with monoclonal gammopathy of undetermined significance, multiple myeloma, Waldenström’s macroglobulinemia/lymphoplasmacytic lymphoma (WM/LPL), cryoglobulinemia, and light-chains amyloidosis [2]. Recently, several studies have suggested that some B-cell lymphomas and chronic lymphocytic leukemia (CLL) may secrete monoclonal proteins and these cases exhibit different clinicopathological characteristics and a more aggressive behavior [3].

In CLL, the prevalence of a serum monoclonal protein, detected by serum immunofixation electrophoresis (sIFE), has been reported to range from 13 to 61% [4,5,6,7,8,9,10,11]. CLL with +sIFE has been associated with advanced stage and adverse biochemical and cytogenetic prognostic features [9,10,11,12]. In addition, a +sIFE has been related to shorter treatment-free [9,10] and overall survival (OS) [8,10,11] in the chemoimmunotherapy era. However, the relationship of this phenomenon with genomic alterations has not been the subject of in-depth research.

Whole genome and exome sequencing data revealed significant molecular heterogeneity in CLL [12,13], which has been confirmed by subsequent studies [14]. In recent years, the cost reduction of next-generation sequencing (NGS) technologies has revolutionized the capacity to characterize the highly complex genetic landscape of CLL [15,16,17]. In this study, we genetically characterize a recent cohort of patients with CLL based on their sIFE, using a custom NGS panel capturing the most common copy number alterations and mutated genes.

## 2. Materials and Methods

### 2.1. Patients

We retrospectively studied 97 patients diagnosed with CLL at a single institution between 2003 and 2023. Peripheral blood samples were collected for DNA extraction from November 2010 to August 2023. The eligibility criteria included the diagnosis of previously untreated CLL based on the standard criteria [18,19,20] and availability of biological material for molecular biology at the same time as sIFE. Clinical, biochemical, and karyotype information was also collected. Patients were treated according to the local policy based on GELLC and ESMO guidelines [21,22]. Informed consent was signed by all patients. The study was approved by the Ethics Committee of Hospital Clínic de Barcelona (ref. number HCB/2023/0079) and was performed in accordance with the Declaration of Helsinki.

### 2.2. Serum Immunofixation Electrophoresis

In all patients, sIFE was performed by electrophoresis (Hydrasys 2, Sebia, Lisses, France) to identify the heavy and light chains involved, at the same time as NGS analysis, always before frontline treatment. A biclonal sIFE was defined as the presence of two heavy and/or light chains on the sIFE. Patients with positive urine immunofixation but negative sIFE were excluded from the study. Concordance between the light chain of the sIFE and the light-chain restriction by flow cytometry of the peripheral blood and/or bone marrow was evaluated.

### 2.3. Next-Generation Sequencing

We used All-CLL, a customized assay from a predesigned SOPHiA GENETICS capture-based NGS panel [18]. All-CLL covers the following genes: *ATM*, *BCL2*, *BIRC3*, *BTK*, *CXCR4*, *EGR2*, *FBXW7*, *KRAS*, *MYD88*, *NFKBIE*, *NOTCH1*, *PLCG2*, *POT1*, *SF3B1*, *TP53*, and *XPO1*, as well as the status of four copy number alterations (CNA) [del(17p13), del(11q22.3), del(13q14.3), and trisomy 12], full-length IGH V(D)J gene rearrangements, and IGHV gene somatic hypermutation. NGS libraries were prepared according to the user guide from 200 ng of genomic DNA and their quality was assessed using the Quality Control Agilent kit (Tape Station, Agilent, Santa Clara, CA; USA and Qubit, Thermo Fisher Scientific, Eugene, OR, USA ). Sequencing was performed using Illumina MiSeq (2 × 300 bp, mean coverage ≈ 1500×). CNAs and gene variants were analyzed with the DDM Platform (Sophia Genetics, Rolle, Switzerland) and immunoglobulin gene rearrangements were reconstructed using IgCaller(v1.3) [23] and analyzed with IMGT/V-QUEST 3.6.3 [24] and ARResT/AssignSubsets 07.01.22 [25], following ERIC recommendations [26].

An analysis filter was applied including a cutoff of >3% of variant allele frequency (VAF), run frequency < 85%, gnomAD < 2%, and a coding consequence: frameshift, inframe, missense, nonsense, no-start, no-stop, splice site, or untranslated regions (UTRs). Accepted gene variants were manually classified as pathogenic, likely pathogenic, or as variants of unknown significance (VUS) using COSMIC (https://Cancer.Sanger.Ac.Uk/cosmic (accessed on 19 September 2024)), VarSome (https://Varsome.Com/ (accessed on 19 September 2024)), Franklin (https://Franklin.Genoox.Com/Clinical-Db/Home (accessed on 19 September 2024)), Seshat for TP53 variants (http://Vps338341.Ovh.Net/ (accessed on 19 September 2024)), and the literature. CNAs detected by NGS and by fluorescent in situ hybridization (FISH) analysis were classified following Döhner’s hierarchical model [27]. Complex karyotype was defined as ≥5 chromosomal abnormalities.

### 2.4. Statistical Analysis

The χ2 or Fisher’s exact tests were used to compare categorical variables, and Student’s t and Mann–Whitney’s U tests were used to compare quantitative variables. We plotted Kaplan–Meier survival curves and used the log-rank test to explore survival differences based on the sIFE. We chose time to first treatment (TTFT, the interval between NGS/sIFE and the initiation of frontline therapy) as the primary endpoint of the study due to its robustness in terms of independence from treatment strategies and causes of death. In order to estimate TTFT, in which death without the primary event is possible, cumulative incidence was calculated (cmprsk package, R software 4.4.1, Vienna, Austria) and compared by use of Gray’s test. The Fine–Gray regression model was used in the multivariable analysis of events with competing risks. To minimize a possible selection bias, survival and TTFT were calculated from the time of NGS. A *p*-value < 0.05 was considered statistically significant.

## 3. Results

### 3.1. Clinical and Laboratory Characteristics

The baseline clinical and laboratory characteristics of the 97 patients with CLL according to the sIFE are shown in Table 1. Forty-nine patients (51%) had a +sIFE. No significant differences were found in the clinical or laboratory characteristics of the patients according to the sIFE.

A mutational analysis by NGS was performed at the time of diagnosis in 59 patients, during follow-up but before treatment in 17 patients and immediately before first-line treatment in 21 patients (median time to treatment, 27 days (range 2–246 days)). Baseline characteristics according to the moment of NGS analysis are shown in Appendix A. The proportion of patients with a +sIFE was similar across all subgroups.

### 3.2. Genomic Characteristics

An overview of the mutated genes, CNAs, and immunoglobulin characteristics according to sIFE is shown in an integrative OncoPlot (Figure 1). The proportion of patients with unmutated IGHV genes was significantly higher among patients with a +sIFE (*n* = 29, 60% vs. *n* = 14, 29%, *p* = 0.004). The IGHV1-69 gene was overrepresented in the +sIFE group (*n* = 9, 19% vs. *n* = 1, 2%, *p* = 0.02), while IGHV4-34 was absent in the +sIFE group (*n* = 0 vs. *n* = 6, 13%, *p* = 0.03, Figure 2). NGS analysis showed that a +sIFE was associated with the presence of mutations in one or more genes (*n* = 32, 65% vs. *n* = 19, 40%, *p* = 0.02). Likewise, the number of mutated genes and the total number of mutations were significantly higher in the +sIFE group [median 2 (range 0–3) vs. 0 (0–2), *p* = 0.006 and median 1 (0–4) vs. 0 (0–3), *p* = 0.007, respectively]. However, no significant differences were found in the mutation rate of any of the genes studied individually, or in CNAs, according to the sIFE (Table 2).

### 3.3. Heavy-Chain Immunoglobulin Isotype

Among the 49 patients with a +sIFE at diagnosis, IgG κ was the most common isotype (27%), followed by bi/triclonal (25%) and λ free light chains (FLC, 20%). The frequency of other isotypes was as follows: IgG λ (10%), IgM κ (10%), IgM λ (4%), κ FLC (2%), and IgA κ (2%). No IgD or IgE cases were identified.

When we analyzed the association between sIFE immunoglobulin isotypes and the presence of mutations in specific genes, an IgM isotype was associated with *TP53* mutations (36% (*n* = 5) for patients with an IgM +sIFE, compared with 12% (*n* = 10) for patients with –sIFE or non-IgM +sIFE, *p* = 0.04). The IgG subtype was associated with SF3B1 mutations (*n* = 5, 18% for IgG +sIFE vs. *n* = 3, 4% for –sIFE or non-IgG +sIFE), while the bi/triclonal immunoglobulin isotypes were associated with *XPO1* and *NOTCH1* mutations (*n* = 3, 25% and *n* = 4, 33% for bi/triclonal sIFE vs. *n* = 3, 4% and *n* = 8, 9% for –sIFE or monoclonal +sIFE, *p* = 0.04). No other associations were found between the heavy-chain isotype and gene mutations (Table 3).

In order to further explore the association between these genomic alterations and the presence of a +sIFE, we extended the analysis to 23 additional previously treated patients for whom we had simultaneous information on sIFE and NGS. A total of 65 patients had a +sIFE at the time of NGS (54%). The isotypes were distributed as follows: IgG 34%, bi/triclonal 28%, FLC 19%, IgM 18%, and IgA 1%. Except for SF3B1 with IgG, the previously described associations between gene mutations and the heavy-chain isotype were maintained: *TP53* mutations were more frequent in IgM +sIFE patients (*n* = 8, 35%) compared with –sIFE or non-IgM +sIFE patients (*n* = 13, 13%), with *p* = 0.04, and the *XPO1* and *NOTCH1* mutations were more frequent in bi/triclonal (*n* = 5, 28% and *n* = 6, 33%, respectively) than in –sIFE or monoclonal +sIFE cases (*n* = 5, 5% and *n* = 10, 10%, respectively) with *p* = 0.04 and *p* = 0.03. No further associations were found.

### 3.4. Light-Chain Immunoglobulin Isotype

The information between the light chain of the serum Ig and the light-chain restriction by flow cytometry of the peripheral blood and/or bone marrow was compared in 46 (94%) cases with a +sIFE: 76% were concordant and 24% were discordant. No association was found between light chain concordance and gene mutations.

### 3.5. Treatment and Survival

Excluding 21 patients whose NGS and sIFE were performed immediately before first-line treatment [median time to treatment, 27 days (range 2–246 days)], the median follow-up was 1.6 years (95% CI 1.5–1.9). Thirty-one patients (41%) received frontline treatment, which is depicted in Table 4. Treatment strategies were comparable between patients with a negative and positive sIFE. The most commonly used therapeutic frontline regimen was BTK inhibitors (94%) with a median treatment duration of 0.7 years (range 0.1–6.7 years). At the end of the follow-up, 43% of the patients had discontinued BTK inhibitor therapy. The main reasons were toxicity (38%) and progression (31%) [Appendix A]. The median TTFT for the cohort was 1.1 years (95% CI 0.7–2 years), and patients with a +sIFE had a significantly shorter TTFT [3-year probability of requiring treatment: +sIFE 54% (24–77%), -sIFE 21% (1–38%); *p* = 0.033] (Figure 3a). Considering the statistically significant factors from the univariable analysis, and omitting factors overlapping with the CLL-IPI, a multivariable model for TTFT was built (Figure 3b and Table 5), in which only the very high-risk CLL-IPI category retained statistical significance.

The median PFS and survival from NGS/sIFE of the cohort were 2.6 years (95% CI 0.7–4.5 years) and 1.6 years (95% CI 1.5–2 years), respectively. No statistically significant differences were seen in PFS or survival from NGS/sIFE according to sIFE (Figure 3c,d), although a trend towards worse outcomes (measured by both endpoints) was seen for patients with a +sIFE. A total of six patients died during follow-up. The causes of death were progression (33%), infection (33%), cardiovascular disease (15%), and second malignancies (15%), with no differences according to sIFE.

## 4. Discussion

Along with other clinical, biochemical, and genetic parameters, the presence of an sIFE has been studied in CLL, with the purpose of deepening the understanding of B-cell biology and eventually tailoring treatment. The prevalence of +sIFE in this cohort was 51%, slightly higher than in other studies, where the data ranged from 13 to 35% [7,8,9], and higher than in our previous publication on this topic [11]. In contrast to our earlier report, +sIFE was not associated with baseline characteristics. A possible explanation for these findings could be a higher tumor burden in this cohort. Indeed, NGS studies are generally performed in patients with higher tumor burden or those in whom treatment is expected soon. For instance, in the present study, the mean absolute lymphocyte count was 34 × 10^9^/L (in contrast to 9 × 10^9^/L in our previous report), and the proportion of patients with CLL-type monoclonal B-cell lymphocytosis (MBL) was only 5% (compared with 20% in our previous study).

In an effort to postulate whether the serum monoclonal was directly secreted by the tumor population, we compared the light chain of the sIFE with the analysis of the light-chain restriction on the surface of the leukemic cells by flow cytometry. We found a concordance of 76%, in the range of those described in previous studies [4,5,6,7,8,9,10,11]. This suggests that the major CLL clone may not be the producer of this protein in all cases.

The analysis of the mutational landscape of CLL has shown an association between the accumulation of driver mutations and a shorter TTFT or worse OS (in the chemoimmunotherapy era), independent of other risk parameters such as Binet stage or IGHV status [28,29,30,31]. In this study, a higher mutational burden was associated with the presence of a +sIFE. We postulate that an increasing mutational burden could somehow lead to the immune dysregulation of the CLL cell population and the secretion of immunoglobulin.

*TP53* alterations are a consistent adverse prognostic marker and a relevant decision factor in therapeutic algorithms [21,32,33]. Interestingly, we found that an IgM +sIFE was associated with all types of *TP53* alterations: *TP53* mutations, del(17p), and *TP53* alterations (*TP53* mutations and/or del(17p)). Previous studies suggested a similar relationship between del(17p) [10] or *TP53* alterations [9] and serum monoclonal IgM. Likewise, bi/triclonal +sIFE were associated with *XPO1* and *NOTCH1* mutations. However, due to the low number of patients, this relationship may be spurious.

In addition, in line with some previous publications [8,11] (but not with some others [7,9,10]), the presence of a +sIFE was associated with an unmutated IGHV status. This relationship was maintained irrespective of the heavy-chain isotype. Nine (69%) IgM and nineteen (67%) IgG +sIFE CLL patients had unmutated IGHV genes. In a mouse model study, Roco et al. established that class-switch recombination occurs infrequently in the germinal center and therefore precedes somatic hypermutation [34]. If this hypothesis were confirmed in humans, this might explain why we detected an IgG +sIFE in some cases with unmutated IGHV.

Previous reports showed an underrepresentation of the IGHV4-34 gene in patients with an IgG +sIFE [9,10]. In the present study, none of the six patients with the IGHV4-34 gene rearrangement had a +sIFE. In addition, similarly to our results (but in contrast with another one [9]), an association between IGHV1-69 and +sIFE has been suggested [10].

A number of limitations of our study need to be taken into account. The relatively small size of the cohort reduced the statistical power for multiple gene comparisons. In addition, the majority of patients included in our series were from 2018 to 2023, when NGS was more widely available. Therefore, the follow-up of the cohort may not have been long enough to identify overall survival differences.

In conclusion, these results point towards an association between the presence of a serum monoclonal component and higher mutational burden, as well as with specific mutations. This could be one of the factors contributing to the negative prognostic value of a +sIFE, although other biological interrelations remain to be identified. A larger validation cohort is needed to further investigate the biological basis of these findings.

## Figures and Tables

**Figure 1 cells-13-01839-f001:**
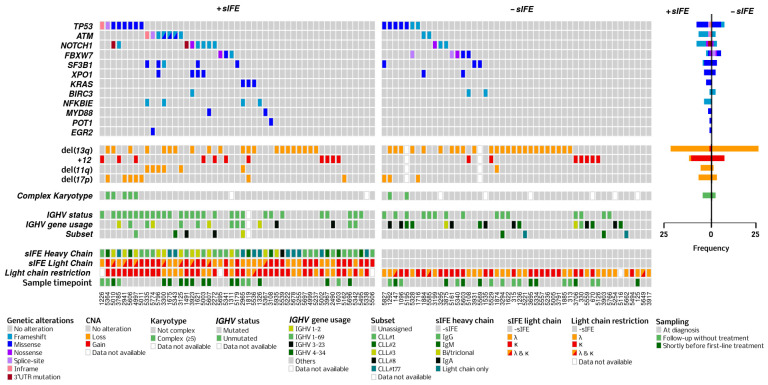
OncoPlot showing NGS results according to the sIFE. Recurrent genomic alterations are ordered by decreasing frequency. Copy number alterations (CNAs) were performed by NGS. Light-chain restriction was established by flow cytometry.

**Figure 2 cells-13-01839-f002:**
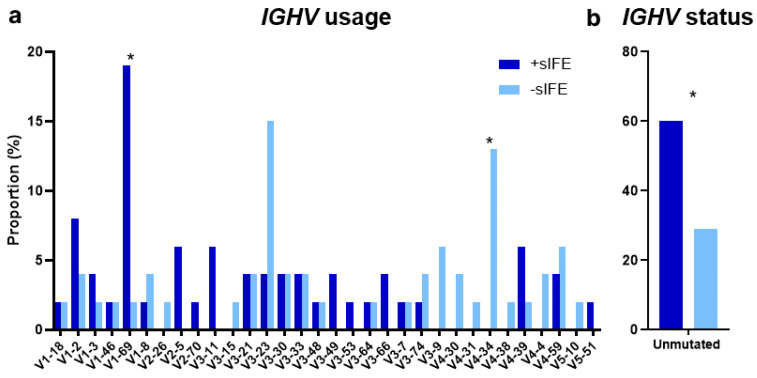
(**a**) IGHV usage according to sIFE. (**b**) IGHV status according to sIFE. * *p* < 0.05.

**Figure 3 cells-13-01839-f003:**
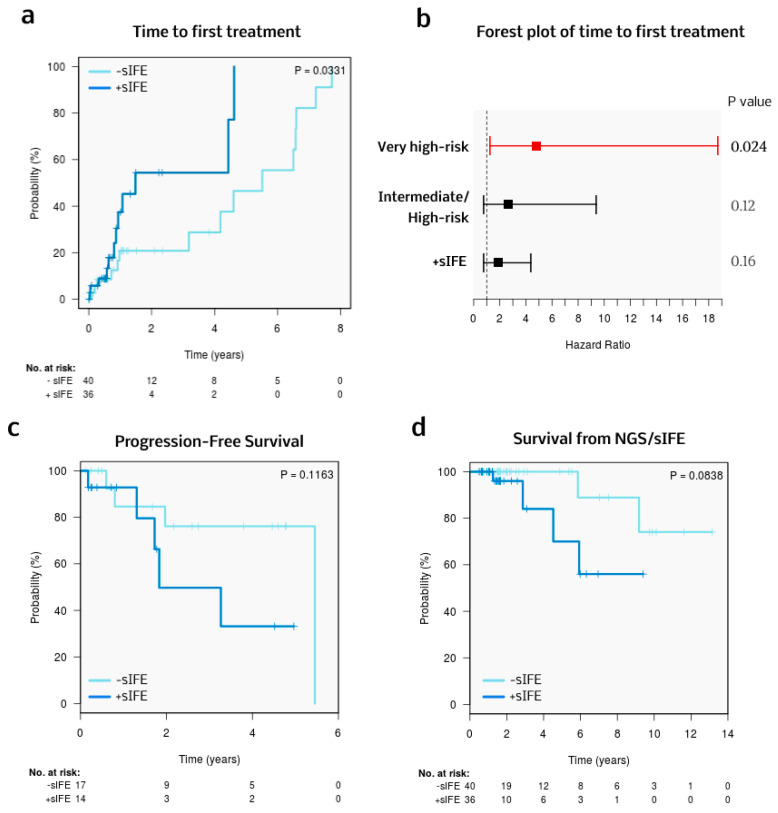
(**a**) Comparison of TTFT between +sIFE and –sIFE cases. (**b**) Forest plot of the multivariable model to predict time to first treatment (CLL-IPI very high-risk, CLL-IPI intermediate/high-risk, and +sIFE). (**c**,**d**) Comparison of PFS and survival from NGS/sIFE between +sIFE and –sIFE.

**Table 1 cells-13-01839-t001:** Baseline clinical and laboratory characteristics according to the sIFE.

		sIFE at the Time of NGS		
Characteristic	All Patients (*n* = 97)	Negative (*n* = 48, 49%)	Positive (*n* = 49, 51%)	*p* Value
Age in years, median (range)	69 (41–96)	68 (44–96)	70 (41–94)	NS
Male sex, *n* (%)	57 (59)	27 (56)	30 (61)	NS
Diagnosis				
CLL, *n* (%)	89 (92)	43 (90)	46 (94)	NS
MBL, *n* (%)	5 (5)	4 (8)	1 (2)	
SLL, *n* (%)	3 (3)	1 (2)	2 (4)	
ECOG PS ≥ 1, *n* (%) (*n* = 89)	8 (9)	2 (4)	6 (14)	NS
B symptoms, *n* (%)	2 (2)	0	2 (4)	NS
Binet stage C, *n* (%)	13 (13)	5 (10)	8 (16)	NS
Rai stage III-IV, *n* (%)	14 (14)	5 (10)	9 (18)	NS
Lymphadenopathy (CT), *n* (%)	35 (36)	14 (30)	21 (43)	NS
ALC > 15 × 109/L, *n* (%)	56 (58)	27 (56)	29 (59)	NS
LDH above ULN, *n* (%) (*n* = 92)	13 (14)	7 (15)	6 (13)	NS
β2-microglobulin above ULN, *n* (%) (*n* = 91)	52 (57)	21 (47)	31 (67)	NS
Complex karyotype (*n* = 88)	7 (8)	2 (5)	5 (11)	NS
FISH, *n* (%)				
Normal	15 (16)	8 (18)	7 (15)	NS
del(13)(q14.3)	37 (40)	22 (49)	15 (32)	
Trisomy 12	13 (14)	5 (11)	8 (17)	
del(11)(q22.3)	8 (9)	2 (4)	6 (13)	
del(17)(p13.1)	19 (21)	8 (18)	11 (23)	
CLL-IPI, *n* (%) (*n* = 87)				
Low	20 (23)	13 (32)	7 (15)	
Intermediate	16 (18)	10 (24)	6 (13)	0.055
High	33 (38)	13 (32)	20 (44)	
Very high	18 (21)	5 (12)	13 (28)	
Diagnosis period				
2003–2007	3 (3)	1 (2)	2 (4)	
2008–2012	13 (13)	9 (19)	4 (8)	NS
2013–2017	19 (20)	10 (21)	9 (19)	
2018–2023	62 (64)	28 (58)	34 (69)	
NGS and sIFE date				
2003–2007	0	0	0	
2008–2012	3 (3)	3 (6)	0	NS
2013–2017	14 (14)	7 (15)	7 (14)	
2018–2023	80 (83)	38 (79)	42 (86)	

ALC, absolute lymphocyte count; ECOG PS, Eastern Cooperative Oncology Group Performance Status; MBL, CLL-type monoclonal B-cell lymphocytosis; NGS, next-generation sequencing; NS, not statistically significant; sIFE, serum immunofixation electrophoresis; SLL, small lymphocytic lymphoma; CLL, chronic lymphocytic leukemia; CT, computed tomography; LDH, lactate dehydrogenase; ULN, upper limit of normal.

**Table 2 cells-13-01839-t002:** Gene mutations by NGS at time of sIFE.

		sIFE at the Time of NGS		
Pathogenic/Likely Pathogenic Mutation	All Patients (*n* = 97)	Negative (*n* = 48, 49%)	Positive (*n* = 49, 51%)	*p* Value
Presence of any mutation, *n* (%)	51 (53)	19 (40)	32 (65)	0.0196
Number of mutations, median (range)	1 (0–4)	0 (0–3)	1 (0–4)	0.0069
Number of mutated genes, median (range)	1 (0–3)	0 (0–2)	1 (0–3)	0.0061
*ATM*, *n* (%)	9 (9)	2 (4)	7 (14)	NS
*BCL2*, *n* (%)	0	0	0	-
*BIRC3*, *n* (%)	3 (5)	2 (4)	1 (2)	NS
*BTK*, *n* (%)	0	0	0	-
*CXCR4*, *n* (%)	0	0	0	-
*EGR2*, *n* (%)	1 (1)	0	1 (2)	-
*FBXW7*, *n* (%)	8 (8)	5 (10)	3 (6)	NS
*KRAS*, *n* (%)	3 (3)	0	3 (6)	NS
*MYD88*, *n* (%)	2 (2)	0	2 (4)	NS
*NFKBIE*, *n* (%)	4 (4)	0	4 (8)	NS
*NOTCH1*, *n* (%)	11 (11)	3 (6)	8 (16)	NS
*PLCG2*, *n* (%)	0	0	0	-
*POT1*, *n* (%)	1 (1)	0	1 (2)	-
*SF3B1*, *n* (%)	8 (8)	3 (6)	5 (10)	NS
*TP53*, *n* (%)	15 (15)	7 (15)	8 (16)	NS
*XPO1*, *n* (%)	6 (6)	2 (4)	4 (8)	NS

NGS, next-generation sequencing; NS, not statistically significant; sIFE, serum immunofixation electrophoresis.

**Table 3 cells-13-01839-t003:** sIFE immunoglobulin isotypes and specific genes mutated.

Mutated Gene, *n* (%)	-sIFE/Non-IgG +sIFE (*n* = 69, 71%)	IgG +sIFE (*n* = 28, 29%)	*p*	-sIFE/Non-IgM +sIFE (*n* = 83, 86%)	IgM +sIFE (*n* = 14, 14%)	*p*	-sIFE/Monoclonal +sIFE (*n* = 85, 88%)	Bi/Triclonal (*n* = 12, 12%)	*p*
*ATM*	4 (6)	5 (18)	NS	7 (8)	2 (14)	NS	6 (7)	3 (25)	NS
*FBXW7*	5 (7)	3 (11)	NS	8 (10)	0	NS	7 (8)	1 (8)	NS
*NFKBIE*	2 (3)	2 (7)	NS	4 (5)	0	NS	3 (4)	1 (8)	NS
*NOTCH1*	6 (7)	6 (21)	NS	10 (12)	2 (14)	NS	8 (9)	4 (33)	0.04
*SF3B1*	3 (4)	5 (18)	0.04	7 (8)	1 (7)	NS	5 (6)	3 (25)	NS
*TP53*	11 (16)	4 (14)	NS	10 (12)	5 (36)	0.04	12 (14)	3 (25)	NS
*XPO1*	2 (3)	4 (14)	NS	5 (6)	1 (7)	NS	3 (4)	3 (25)	0.03

Only genes with more than 3 mutated cases are shown. sIFE, serum immunofixation electrophoresis.

**Table 4 cells-13-01839-t004:** Treatments according to sIFE.

		sIFE at the Time of NGS		
First Line of Treatment *	Patients (*n* = 76)	Negative (*n* = 40, 53%)	Positive (*n* = 36, 47%)	*p* Value
Untreated, *n* (%)	45 (60)	24 (60)	21 (58)	NS
BTKi, *n* (%)	29 (38)	15 (37)	14 (39)	
Ibrutinib	20 (70)	11 (74)	9 (65)	
Acalabrutinib	5 (17)	2 (13)	3 (21)	
Zanubrutinib	4 (13)	2 (13)	2 (14)	
BCL2i, *n* (%)	1 (1)	0	1 (3)	
Immunochemotherapy *n* (%)	1 (1)	1 (3)	0	

BCL2i, B-cell lymphoma 2 inhibitor; BTKi, Bruton tyrosine kinase inhibitors. * Always after sIFE and NGS.

**Table 5 cells-13-01839-t005:** Predictors of time to first treatment in the univariable and multivariable analyses.

		Univariable Analysis		Multivariable Analysis (75 Cases, 27 Events)	
Parameter	Risk Category	HR (95% CI)	*p*	HR (95% CI)	*p* Value
Sex	Male	0.72 (0.34–1.53)	NS	NI ^a^	
Age	>65 years	1.44 (0.63–3.32)	NS	NI ^b^	
Stage	Binet B-C and/or Rai I-IV	3.47 (1.39–8.68)	0.008	NI ^b^	
ECOG PS	≥1	1.85 (0.47–7.27)	NS	NI ^a^	
ALC	>15 × 10^9^/L	1.02 (0.49–2.13)	NS	NI ^a^	
B2M levels	>3.5 mg/L	3.13 (1.33–7.4)	0.01	NI ^b^	
IGHV status	Unmutated	4.16 (1.69–10.25)	0.001	NI ^b^	
del(17p) and/or TP53 mutation	Present	1.28 (0.56–2.9)	NS	NI ^b^	
CLL-IPI risk group	Intermediate/ High-risk	2.98 (0.85–10.39)	NS	2.68 (0.76–9.4)	NS
	Very high-risk	6.28 (1.68–23.52)	0.006	4.8 (1.32–18.71)	0.02
sIFE	Positive	2.6 (1.14–6.08)	0.03	1.86 (0.79–4.41)	NS

ALC, absolute lymphocyte count; B2M, β2-microglobulin; CI, confidence interval; ECOG PS, Eastern Cooperative Oncology Group Performance Status; HR, hazard ratio; LDH, lactate dehydrogenase; NI, not included; NS, not statistically significant; ULN, upper limit of normal; ^a^, not included due to absence of statistical significance in the univariable analysis; ^b^, not included to avoid overlapping with CLL-IPI.

## Data Availability

The authors confirm that the data supporting the findings of this study are available within the article and its Appendix A.

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
