# Peer review of "Association of Genomic Alterations with the Presence of Serum Monoclonal Proteins in Chronic Lymphocytic Leukemia"

_cells, 2024, doi:10.3390/cells13221839_

Round 1

Reviewer 1 Report

Comments and Suggestions for Authors

The introduction is appropriate highlighting the problem of sIFE in CLL and provides the problem of sIFE in CLL in terms of clinical manifestation and the course of disease. Methods are clearly stated and follow the inner logic of the study. My only objection is that the authors used the TTTF as the primary endpoint without explaining it fully and the reasons why this particular endpoint is taken. Results are clearly written.I suggest that instead in the supplementary file the univariate and multivariate analysis to be included in this section. Discussion is adequate, but authors should stress out the possible limitations of the study.   The introduction is appropriate highlighting the problem of sIFE in CLL and provides the problem of sIFE in CLL in terms of clinical manifestation and the course of disease. Methods are clearly stated and follow the inner logic of the study. My only objection is that the authors used the TTTF as the primary endpoint without explaining it fully and the reasons why this particular endpoint is taken. Results are clearly written.I suggest that instead in the supplementary file the univariate and multivariate analysis to be included in this section. Discussion is adequate, but authors should stress out the possible limitations of the study.   

Reviewer 2 Report

Comments and Suggestions for Authors

 The paper by Pineyroa and colleagues finds merit as the literature on the topic is scarce, especially regarding evaluation in NGS. However, some issues preclude manuscript publication in its current form

Minor issues

Abstract, lines 6-7, the sentence about gene mutation is not clear, especially the distribution of brackets, please clarify.

How was the diagnosis of CLL made, with MAtutes score? via Hallek 1996 and/or 2008 CLL giudelines?

Was the informed consent signed? And for patients who died or lost at follow up?

How was the treatment regimen chosen? Hallek 2008, ESMO,NCCN giudelines?

CLL IPI should be included in Table 1

3.5 Median TTFT fot sIPE+ and - should be reported

As BTK inhibitors, was ibrutinib used? Or acalabrutinib/zanubrutinib? Please specify.

MAjor issues

The case series is perhaps not representative of the entire population with CLL. In fact, the diagnosis started in 2003 but no one was treated before 2013. How is it possible to have a TTFT of 1.1 years with cases also diagnosed in 2003? It would be useful to divide the year of diagnosis in Table 1, at least divided into 5-year periods (2003-2007, 2008-2012, 2013-2017, 2018-)
Treatment dosage, response, treatment duration, toxicity should be reported. In how many patients is treatment still ongoing? Was monoclonal component reduced or disappeared during treatment for responsive patients?

In a separate paragraph, PFS and OS should be carefully described, with median survival, confidence interval. The difference between 2 groups, sIFE+ and -,  should be reported as hazard ratio with p value in the text.

Causes of death should be reported.

Round 2

Reviewer 2 Report

Comments and Suggestions for Authors

DEar Author,
The manuscript was largerly improved.
The bias regarding TTFT, which considers The execution of the sampling for NGS as T0, remains. I think it is not possible to draw any conclusion and the sentence should be removed in the abstract.
The authors say t
he median PFS and OS were 2.6 and 1.6 years. How is it possible that PFS is prolonged if compared to OS? Please verify.
